# GraphATT-DTA: Attention-Based Novel Representation of Interaction to Predict Drug-Target Binding Affinity

**DOI:** 10.3390/biomedicines11010067

**Published:** 2022-12-27

**Authors:** Haelee Bae, Hojung Nam

**Affiliations:** 1AI Graduate School, Gwangju Institute of Science and Technology, 123 Cheomdangwagi-ro, Buk-gu, Gwangju 61005, Republic of Korea; 2School of Electrical Engineering and Computer Science, Gwangju Institute of Science and Technology, 123 Cheomdangwagi-ro, Buk-gu, Gwangju 61005, Republic of Korea; 3Center for AI-Applied High Efficiency Drug Discovery (AHEDD), Gwangju Institute of Science and Technology, 123 Cheomdangwagi-ro, Buk-gu, Gwangju 61005, Republic of Korea

**Keywords:** deep learning, drug-target interaction, binding affinity, graph neural network, attention

## Abstract

Drug-target binding affinity (DTA) prediction is an essential step in drug discovery. Drug-target protein binding occurs at specific regions between the protein and drug, rather than the entire protein and drug. However, existing deep-learning DTA prediction methods do not consider the interactions between drug substructures and protein sub-sequences. This work proposes GraphATT-DTA, a DTA prediction model that constructs the essential regions for determining interaction affinity between compounds and proteins, modeled with an attention mechanism for interpretability. We make the model consider the local-to-global interactions with the attention mechanism between compound and protein. As a result, GraphATT-DTA shows an improved prediction of DTA performance and interpretability compared with state-of-the-art models. The model is trained and evaluated with the Davis dataset, the human kinase dataset; an external evaluation is achieved with the independently proposed human kinase dataset from the BindingDB dataset.

## 1. Introduction

Drug development is a high-risk industry involving complex experiments, drug discovery, and pre-clinical and clinical trials. Drug discovery is the process of identifying new candidate compounds with potential therapeutic effects, and it is essential for identify drug-target interactions (DTIs). Moreover, the drug-target binding affinity (DTA) provides information on the strength of the interaction between a drug-target pair. However, as there are millions of drug-like compounds, it can take years and costs about 24 million US dollars for experimental assays of target-to-hit process for a new drug [1]. Efficient computational models for predicting DTA are urgently needed to speed up drug development and reduce resource consumption.

There are several computational approaches to predicting DTA [2,3]. One is the ligand-based method, which compares a query ligand to known ligands based on their target proteins. However, prediction results become unreliable if the number of known ligands with target proteins is insufficient [4]. Another approach is molecular docking [5], which simulates the binding of the conformational spaces between compounds and proteins based on their three-dimensional (3D) structures. However, it is too challenging to produce the 3D protein-ligand complex. Another approach is the chemogenomic method [6] that integrates the chemical attributes of drug compounds, the genomic attributes of proteins, and their interactions into a unified math framework.

In feature-based chemogenomic methods, drug-target pairs are taken as input, and their binding strength or whether to interact, determined by regression or binary classification, are output [7,8]. Efficient input representation is key to accurate prediction. The commonly used drug descriptor is chemical fingerprints such as Extended Connectivity Fingerprint [9] or Molecular ACCess System [10]. The commonly used protein descriptor is physicochemical properties, such as amino acid composition, transition, and distribution. On constructed features, random forest, support vector machine, and artificial neural network models are applied to predict these interactions [11]. Similarity information is also used for representation [12,13,14,15]. KronRLS [14] constructs the similarity between drugs or between target proteins with compound similarity and Smith-Waterman similarity. SimBoost [15] constructs features for each drug, target, and drug-target pair from the similarity. However, the fixed lengths of manually selected features may result in the loss of information.

Recently, data-driven features learned during training using large datasets have been shown to increase DTI prediction performance [16,17,18,19,20,21,22,23,24]. DeepDTA learns the representation of drugs and proteins with one-dimensional (1D) convolutional neural network (CNN). However, this leaves the molecule’s original graph structure unaddressed. To cover this, GraphDTA [17] represents a molecule as a graph in which a node is an atom, and an edge is a bond. Graph neural networks (GNN) are used for molecular representation and 1D CNNs are used for protein representation. Additionally, DGraphDTA [18] represents a protein as a contact map followed by graph convolutional network (GCN) embedding to learn DTA using protein structure. However, when modeling DTA interactions, these models consider only the global interactions between compounds and proteins.

Furthermore, several studies [20,21,22,23,24] have introduced attention mechanisms to better model the interactions between drugs and proteins for DTA prediction. DeepAffinity [20] introduced an attention mechanism used to interpret predictions by isolating the main contributors of molecular fragments into their pairs. ML-DTI [21] propose the mutual learning mechanism. It takes input as Simplified Molecular-Input Line-Entry System (SMILES) and amino acid sequences, and 1D CNNs are used for encoding. It leverages protein information during compound encoding and compound information during protein encoding, resulting in a probability map between a global protein descriptor and a drug string feature vector. MATT-DTI [22] proposes a relation-aware self-attention block to remodel drugs from SMILES data, considering the correlations between atoms. With this, 1D CNN is used for encoding. The interaction is modeled via multi-head attention, in which the drug is regarded as a key and the protein as a query and value. HyperAttentionDTI [23] uses a hyperattention module that models semantic interdependencies in spatial and channel dimensions between drug and protein sub-sequences. FusionDTA [24] applies a fusion layer comprising multi-head linear attention to focus on important tokens from the entire biological sequence. Additionally, the protein token is pre-trained with a transformer and encoded by bidirectional long short-term memory (BI-LSTM) layers. 

Although these studies successfully apply attention mechanisms for DTA prediction, they are limited because they learn from less informative input features that do not consider the essential regions needed to determine interaction affinities [16,20,21,22,23,24]. Therefore, in this paper, we propose GraphATT-DTA, an attention-based drug and protein representation neural network that considers local-to-global interactions for DTA prediction (Figure 1). The molecular graph of the compound and protein amino acid sequences are the initial inputs. A powerful GNN model is used for compound representation, and 1D CNNs are used for protein representation. The interactions between compounds and proteins are modeled with an attention mechanism by capturing the important subregions (i.e., substructures and sub-sequences) so that the fully connected layer can predict the binding affinity between a compound and its target protein. We evaluate the performance of our model using the Davis kinase binding affinity dataset and the public, web-accessible BindingDB database of measured binding affinities. GraphATT-DTA’s prediction performance is then compared with state-of-the-art (SOTA) global and local interaction modeling methods.

## 2. Materials and Methods

### 2.1. Dataset

In this study, our proposed model and comparison baselines were trained with the Davis dataset [25] and evaluated for external validity with the BindingDB dataset [26]. Table 1 and Appendix A provide a summary. The Davis dataset contains the kinase protein family and relevant inhibitors with dissociation constants $K_{d}$, whose value is transformed into the log space as
(1)$$pK_{d}=-\log_{10}\left(\frac{K_{d}}{e^{9}}\right)$$

The BindingDB dataset is publicly accessible and contains experimentally measured binding affinities whose values are expressed as $K_{d},\;\;K_{i},\;\mathrm{IC}50,\;\mathrm{and}\;\mathrm{EC}50$ terms. For the external test, we extracted drug-target pairs in which the protein is human kinase, and the binding affinity is recorded as a $K_{d}$ value. These values are then transformed into the log space as described.

The Davis dataset consists of six parts. Five are used for cross-validation and one is used for testing. We use the same training and testing scheme as GraphDTA. The hyperparameter is tuned using five parts with five-fold cross-validation. After tuning the hyperparameter, we train all five parts and evaluate the performance with one test part. To evaluate the generalizability of the model, BindingDB is used as the external test dataset. 

### 2.2. Input Data Representation

GraphATT-DTA takes SMILES as the compound input and amino acid sequence string as the protein input. First, the SMILES string is converted to a graph structure that takes atoms as nodes and bonds as edges using the open-source Deep Graph Library (DGL) v.0.4.3(2) [27], DGL-LifeSci v.0.2.4 [28], and RDKit v.2019.03.1(1) [29]. We used the atomic feature defined in GraphDTA (i.e., atom symbol, number of adjacent atoms, number of adjacent hydrogens, implicit values of the atoms, and whether the atom is in aromatic structures). We leverage the bond feature used by the directed message-passing neural network (DMPNN; i.e., bond type, conjugation, in the ring, stereo). Table 2 and Table 3 list detailed information for each feature. Each amino acid sequence type is encoded with an integer and cut by a maximum length of 1000. If the sequences are shorter than the maximum length, they are padded with zeros. The maximum length can cover at least 80% of all proteins.

### 2.3. Drug Representation Learning Model

The molecule is originally represented by a graph structure consisting of atoms and bonds. The GNN uses its structural information and applies a message-passing phase consisting of message_passing and update functions. In the message_passing function, node v aggregates information from its neighbor’s hidden representation, hw(t). In the update function, it updates the previous hidden representation, hv(t), to a new hidden representation, hv(t+1), using messages mv(t+1) and the previous step of hidden representation, hv(t):(2)mv(t+1)= message_passing({hw(t), ⋁w∈N(v)})
(3)hv(t+1)=update(mv(t+1), hv(t))
where N(v) is the set of the neighbors of v in graph G, and hv(t) follows time step t of initial atom features, xv. This mechanism, in which atoms aggregate and update information from neighbor nodes, captures information about the substructure of the molecule. GNN models have variants, such as the graph convolutional network (GCN) [30], graph attention network (GAT) [31], graph isomorphism network (GIN) [32], message-passing neural network (MPNN) [33], and directed message-passing neural network (DMPNN) [34], which can be leveraged by specifying the message_passing function, mv(t+1), and update function, hv(t+1) (see Table 4). The output is a drug embedding matrix, D∈ℝNa×d, where Na is the number of atoms, and d is the dimension of the embedding vectors. In the drug embedding matrix, each atom has the information of its neighbor atoms (i.e., substructure) along with the number of GNN layers.

### 2.4. Protein Representation Learning Model

The Davis and BindingDB datasets have 21 and 20 amino acid types, respectively. Hence, we consider 21 and 20 amino acids for learning and testing, respectively. The integer forms of protein amino acid sequences become the input to the embedding layers. These are then used as input to three consecutive 1D convolutional layers, which learn representations from the raw sequence data of proteins. The CNN models capture local dependencies by sliding the input features with filters, and their output is the protein sub-sequence embedding matrix, S∈ℝNs×d , where Ns is the number of sub-sequences. The number of amino acids in a sub-sequence depends on the filter size. The larger the filter size, the greater the number of amino acids in the sub-sequence.

### 2.5. Interaction Learning Model

The relationship between the protein and the compound is a determinant key for DTA prediction. The attention mechanism can make the input pair information influence the computation of each other’s representation. The input pairs can jointly learn a relationship. GraphATT-DTA model constructs the relation matrix R using dot product of protein and compound embedding where R∈ℝNa×Ns. It provides information about the relationship between the substructures of compounds and protein sub-sequences.
(4)R=(D⋅ST)

GraphATT-DTA reflects the local interactions by considering the crucial relationships between protein sub-sequences and compound substructures. The subseq-wise/atom-wise SoftMax is applied to the relation matrix to construct the substructure and sub-sequence significance matrices. The formulas appear in (5) and (6). The element of substructure_significance indicates the substructure’s importance to the sub-sequence. Similarly, the element of subsequence_significance indicates the sub-sequence’s importance to the substructure.
(5)$$\mathrm{substructure}\_\mathrm{significance}=a_{\mathrm{ij}}=\frac{\exp\left(r_{\mathrm{ij}}\right)}{\sum_{i=1}^{N_{a}}\exp\left(r_{\mathrm{ij}}\right)}$$
(6)$$\mathrm{subsequence}\_\mathrm{significance}=s_{\mathrm{ij}}=\frac{\exp\left(r_{\mathrm{ij}}\right)}{\sum_{j=1}^{N_{s}}\exp\left(r_{\mathrm{ij}}\right)}\;$$

The substructure_significance is directed to the drug embedding matrix via element-wise multiplication (⊙) with aj and D, where aj∈ℝNa×1, and j=1, …, Ns. aj indicates each substructure’s importance of the *j*th sub-sequence. D(j)′∈ℝNa×d indicates the drug embedding matrix with the importance of the *j*th sub-sequence.
(7)D(j)′=aj⊙D

Drug vector d(j)″ is constructed by (8) and carries the information of the compound and the *j*th sub-sequence, where d(j)″ ∈ℝ1×d.
(8)d(j)″=∑aD(j)ab′
(9)D″=concat[d(1)″,d(2)″, …, d(Ns)″]

The concatenation of d(j)″ with all sub-sequences causes D″ to inherit all information about the sub-sequences and compounds, where D″∈ℝNs×d. The new drug feature is thus constructed to reflect all protein sequences and compound atoms where drug_feature∈ℝ1×d.
(10)drug_feature=∑iDij″

The new protein feature is calculated the same way. Using the element-wise multiplication of the subsequence_significance and protein embedding matrix, the protein embedding matrix, P(i)′, with the *i*th substructure significance to the sub-sequence, is constructed so that si∈ℝ1×Ns, S∈ℝNs×d , and P(i)′∈ℝNs×d. The summation of P(i)′ makes protein vector p(i)″ with the sub-sequence information about the compound, where p(i)″∈ℝ1×d. After the concatenation of p(i)″, the summation of P″ makes the new protein feature vector reflect compound sub-structure significance information, where P″∈ℝNa×d, and protein_feature∈ℝ1×d.
(11)P(i)′=si⊙ST
(12)p(i)″=∑aP(i)ab′
(13)P″=concat[p(1)″, p(2)″, …, p(Na)″]
(14)protein_feature=∑iPij″ 

Protein and drug features reflecting the local-to-global interaction information are collected via concatenation. The fully connected layers can then predict the binding affinity. We use mean squared error (MSE) as the loss function.

### 2.6. Implementation and Hyperparameter Settings

GraphATT-DTA was implemented with Pytorch 1.5.0 [35], and the GNN models were built with DGL v.0.4.3(2) [27] and DGL-LifeSci v.0.2.4 [28]. Early stopping was configured with the patience of 30 epochs to avoid potential overfitting and obtain improved generalization performance. The hyperparameter settings are summarized in Table 5. Multiple experiments are used with five-fold cross-validation, applied for hyperparameter selection. 

The layers of GNN are important because they pertain to how many neighbor nodes are regarded by the model. Because there are many layers, the model can consider many neighbors; however, doing so can cause an over-smoothing problem in which all node embeddings converge to the same value. Additionally, if the number of layers is too small, the graph substructure will not be captured. Therefore, proper layer configuration is important. The optimal number of GNN layers was experimentally chosen for GraphATT-DTA by using each GNN graph embedding model. Specific experimental results can be found in Appendix A Appendix A.

## 3. Results

### 3.1. Performance Evaluation Metrics

We used the concordance index (CI) and MSE to evaluate prediction performance. We formulated the CI to estimate whether the predicted binding affinity values were in the same order as their true values. $b_{x}$ is the prediction value with the larger affinity, $d_{x}$, $b_{y}$ is the prediction value with the smaller affinity, $d_{y}$, Z is a normalization constant, and *h*(*x*) is the step function.
(15)$$\mathrm{CI}=\frac{1}{Z}\underset{d_{x}>d_{y}}{\sum }h\left(b_{x}-b_{y}\right)$$
(16)$$h\left(x\right)=\{\begin{array}{cc} 1,\;\; & if\;x>0\\ 0.5,\;\; & if\;x=0\\ 0,\;\; & if\;x<0 \end{array}$$

The MSE was used to calculate the differences between predicted and true affinity values, $p_{i}$ is the predictive score, $y_{i}$ is the ground-truth score, and n is the number of samples.
(17)$$\mathrm{MSE}=\frac{1}{n}\overset{n}{\underset{i=1}{\sum }}{\left(p_{i}-y_{i}\right)}^{2}$$

### 3.2. Binding Affinity Prediction on Testing Data

First, we investigated the predictive performance of GraphATT-DTA using various graph embedding models. The drug was represented using different graphs embedding models such as GCN, GAT, GIN, MPNN, and DMPNN. We performed ten times repeated training and testing. In Figure 2, we report the mean and standard deviation scores of the MSE and CI from the Davis dataset. The drug embedding models showed similar performances (i.e., MSE and CI). For MSE, the GIN drug embedding model performed best with an MSE of 0.213, followed by the GCN and MPNN models with MSEs of 0.214 and 0.215. For CI, MPNN performed best, with a CI of 0.899, followed by DMPNN and GCN.

Next, using the compound embedding model, we evaluated the overall affinity prediction performance again using the Davis dataset. Figure 3 illustrates the correlations between the true and predicted affinity scores. Here, MSE, Pearson correlation, CI, and R^2^ values are reported on graphs’ *y*-axes. When the model predicts the binding affinity perfectly, the slope follows the *y* = *x* line. In the Davis dataset, we observed that the predicted distribution follows the true binding affinity quite well (MSE 0.204).

### 3.3. Comparison with Other Methods 

For comparisons, we considered both global and local interaction modeling methods. Global methods included DeepDTA and GraphDTA, and local methods included DeepAffinity, ML-DTI, HyperAttentionDTI, and FusionDTA. For a fair comparison, we used the same training and testing datasets. The Davis dataset consists of six parts: five for training and one for testing. Finally, the model showing the best prediction performance was selected, and if the model used early stopping for training, we also applied early stopping on the Davis testing dataset to avoid overfitting. We trained using the same hyperparameter settings described in the baseline model papers [16,17,20,21,23,24]. 

**DeepDTA** used 1D CNNs to learn representations from the raw sequence data from SMILES of drugs and amino acids of a protein. After pooling the final convolution layer, the drug and protein features were concatenated, and the fully connected layers regressed the DTA.

**GraphDTA** represents drugs as molecule graphs and uses GNNs for molecule representation. The amino acid sequences were the protein inputs, and 1D CNNs were used for protein representation. Max-pooling was applied, followed by concatenation, and the DTA was predicted using hidden layers. GCN, GAT, GIN, and GCN–GAT models were tested, and GIN was the best performer. Hence, we used it for molecule embedding. It did not use early stopping; thus, the best model performance on the Davis testing dataset was selected for testing on the BindingDB.

**DeepAffinity** [19] takes the SMILES string of the compound and the structural propery sequence (SPS) of the protein. SPS includes secondary structure elements, solvent accessibility, physico-chemical characteristics, and lengths. The secondary structure element and solvent accessibility are determined using the SSPro prediction model [31], and the input is pretrained using the Seq2seq autoencoder. After the recurrent neural network (RNN) encodes the input, the pair is represented by a weighted compound and protein summation. Joint attention is then calculated using the compound and protein string pairs. In the original study, the researchers predicted the pIC50 value using a model trained by the BindingDB dataset.

**ML-DTI** [20] applies a mutual learning mechanism and takes SMILES and amino acid sequences as input; 1D CNNs are used for encoding. The model leverages protein information during compound encoding and vice versa. It creates a probability map between the global protein descriptor and drug string feature vector. In the original study, the researchers used the Davis dataset processed by PADME for training and testing. The number of drugs was 72, the targets were 442, and the interactions were 31,824. However, to compare the performances, we used the same Davis dataset as GraphATT-DTA, which consisted of 68 drugs and 442 targets. 

**HyperAttentionDTI** takes compound SMILES and protein input as amino acid sequences. 1D CNN layers encode the compound and protein representations, and a hyperattention module models the semantic interdependencies spatially and channel-wise between the drug and protein sub-sequences. DTI is predicted via binary classification; hence, we changed the last layer and applied MSE to the loss function. 

**FusionDTA** uses a pretrained transformer and BI-LSTM to encode amino acid sequence, and BI-LSTM to encode SMILES. The original researchers proposed a fusion layer, which consisted of a multi-head linear attention layer that focused on the important token from the biological sequence and aggregated global information based on the attention score. The fully connected layer then predicts binding affinity. 

Table 6 and Table 7 report the MSE and CI scores of the Davis and external BindingDB datasets, respectively. We used the model trained on the Davis dataset to evaluate the BindingDB external dataset and verify its generalizability. GraphATT-DTA consistently performed well on both sets. 

For the Davis dataset, the results are shown in Table 6. GraphATT-DTA achieved an MSE of 0.204 and a CI of 0.904. The local interaction modeling methods tend to exhibit superior performance to the global interaction methods. Local interaction methods employ local characteristics extracted from protein and compound to predict DTA. This gives the model a more comprehensive insight into the interacting parts of the protein and the compound. Thus, such models perform better when predicting drug-target affinity based on the training data.

For the external BindingDB dataset test, GraphATT-DTA achieved an MSE of 1.582 and a CI of 0.651. The BindingDB dataset consisted of various large human kinase DTA pairs. The local interaction modeling methods had better results because the attention mechanism caused the model to have a more informative representation. Additionally, early stopping seems to have played an important role as it avoided overfitting. Compared with other models, GraphATT-DTA showed consistently higher performance on both the Davis and BindingDB datasets. Additionally, when comparing the performance per drug scaffold, we confirmed that no particular prediction model has a specially high or low prediction performance for a specific scaffold (Appendix A [36]). However, there were performance gaps between the two. We speculate that discrepancies in the predicted affinity values vs. actual affinity values (pKd) in the two datasets led to performance deterioration. That is, in the Davis dataset, the lowest pKd value was five (10 µM). However, for the BindingDB dataset, many samples had values lower than five (Appendix A). 

### 3.4. Ablation Study 

Next, we performed an ablation study of the GraphATT-DTA model to confirm the advantages obtained by the attention interaction. The ablation model used max-pooling on the drug and protein embedding matrices, followed by concatenation. The fully connected layers predicted the binding affinity. The drug and protein embedding modules were the same as those in the proposed GraphATT-DTA model. Figure 4 compares the performances of concatenation and attention. The results showed that interaction modeling with the attention mechanism had more predictive power.

### 3.5. Visualization with a Case Study

The proposed GraphATT-DTA uses an attention mechanism to consider the important interactions between molecule substructures and protein sub-sequences when predicting the DTA score. For substructure and sub-sequence significance score matrices, we investigated where high attention scores existed. Thus, we prepared a visualized case study experiment for the GraphATT-DTA (Figure 5, Appendix A). For this, we selected the complex compound Lapatinib and gene EGFR (PDB ID: 1XKK). The actual affinity was 8.38, and the predicted affinity was 7.58 via GraphATT-DTA using GCN.

For the 1XKK complex, we found that the high attention scores were clustered near the residue, from 780 to 807 of the protein and quinazoline substructure of the drug. The binding positions of the RCSB PDB showed that the focused interaction between the sub-sequence and substructure, as determined by GraphATT-DTA was one of the true binding sites. The residue, 793 Methionine, and atom, Nitrogen, have a true hydrogen bond. Figure 5a shows the 3D structure of Lapatinib and EGFR, where we colored a ligand as green, the true binding sites are yellow, and hydrogen bonds are red. In Figure 5b, we colored ligand substructures with a high attention score in orange and protein regions with a high attention score in light blue.

The substructure significance is visualized in Figure 5b, where a high score was matched to atoms. From the drug substructure’s perspective, the quinazoline substructure referred to the sub-sequence index, including 793 methionine as the important protein sub-sequence. Figure 5c shows the significances matched to the sub-sequences. From the protein sub-sequence’s perspective, the sub-sequence index, including 793 methionine, regarded quinazoline as the important compound substructure. We identified the top five significance scores from the two significance matrices, finding that they had the same substructure-subsequence pairs. This result indicates that the sub-sequence deemed important by the substructure and the substructure deemed important by the sub-sequence retain the same position in the drug target pair.

## 4. Discussion and Conclusions

Identification of DTIs is an essential step for finding novel drug candidates for a given target protein or vice versa. Once initial binding molecules are successfully identified, the next step is to optimize the activity by enhancing the binding affinity between molecules. Thus, DTA prediction models could be used in many efficient ways in the compound optimization process.

In this paper, we proposed an attention-based novel representation that considers local-to-global interactions to predict DTA. We used a powerful GNN to describe the raw graph data of drugs and 1D CNNs for raw amino acid sequences of proteins. The attention mechanism was used to construct a new molecule and the protein feature that reflects the relationship between the compound substructure and protein sub-sequence. Consequently, the physical interaction patterns between proteins and compounds can be captured by these attention-based protein sequential and molecule graph feature representations. Moreover, in our model, binding patterns of proteins with diverse 3D structures can be addressed during the learning phase and it is one of the advantages of employing 2D sequential features as opposed to 3D structural features. In this regard, the proposed method is not limited to be applied to specific structured proteins but can be applied to more generally structured proteins.

In the case of the performance evaluation, prediction results with the Davis and BindingDB datasets clearly demonstrated the efficacy of the proposed method. We observed that the precise modeling of the interaction allowed for more efficient feature learning from training data. Consequently, when modeling DTA, if interaction patterns are observed during training for affinity prediction, the prediction performance can be improved to the point where the model can provide guidance regarding unidentified drug-protein interaction regions. In addition, our case study demonstrated that GraphATT-DTA can provide unique biological insights to help understand the predicted binding affinity based on the attention scores. We also identified a substructure with high attention scores of a compound that triggers decreasing the binding affinity between EGFR and Canertinib (Appendix A [37]). Notably, it is well known that the overall performance of the deep learning model heavily depends on the training data. Due to the fact that only kinase inhibitors are included in the training dataset used in this study, the performance of the proposed model against other proteins may be compromised. Thus, in order to provide more general DTA predictions, it is necessary to construct large-scale benchmark datasets containing different classes of proteins.

The mutation of protein can lead to human disease. The mutated amino acid can change biological functions and three-dimensional structures that can change the binding affinity. However, no SOTA models have yet incorporated the concept of mutational alteration effects on protein sequence for modeling DTA (Appendix A). For future research, it is necessary to develop a deep-learning model that can identify small-scale alterations in a protein sequence and apply their effects to predicting DTIs and DTAs. 

Recently, deep-learning models that predict protein structures based only on amino acid sequences have been developed [38,39]. With such methods, the binding pockets of proteins can be captured, and interactions can be modeled for drug target affinity prediction. We believe that such approaches will soon allow us to achieve higher performance, and leave the exploration using interaction sites with binding pockets in a data-driven manner to future work.

## Figures and Tables

**Figure 1 biomedicines-11-00067-f001:**
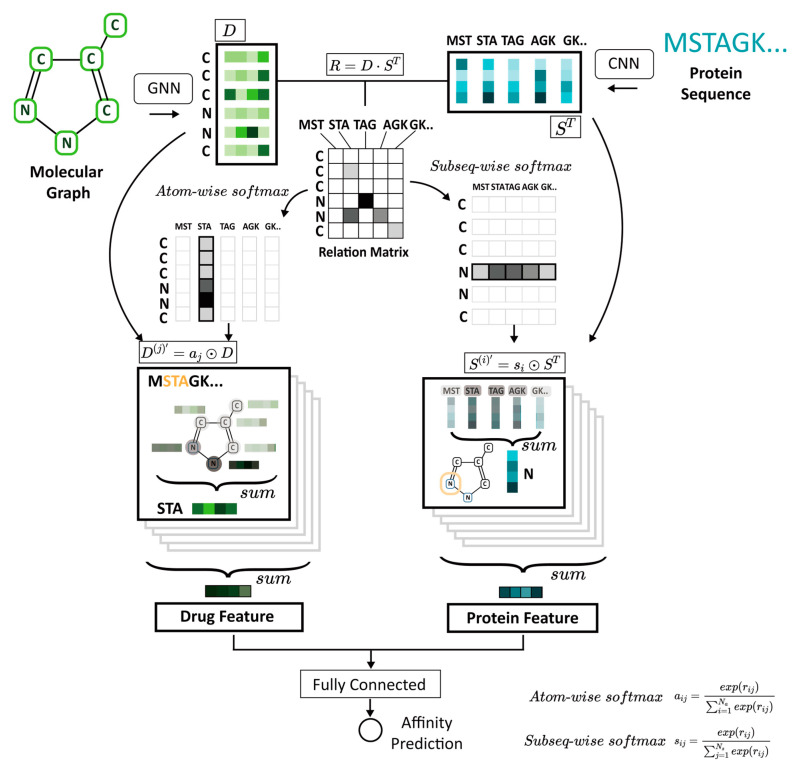
GraphATT-DTA architecture. Molecule graph and protein amino acid sequences are taken as input, and the predicted binding affinity of drug-target pair is the output. The attention-based novel representation is learned by molecule encoding with a graph neural network, sequence encoding using convolutional neural networks, and interaction modeling. The local-to-global relationships between drug substructures and protein sub-sequences are learned via interaction modeling using an attention mechanism. CNN, convolutional neural network; GNN, graph neural network; D, drug embedding matrix; S, protein embedding matrix; R, relation matrix; a, atom-wise softmax of relation matrix; s, subseq-wise softmax of relation matrix.

**Figure 2 biomedicines-11-00067-f002:**
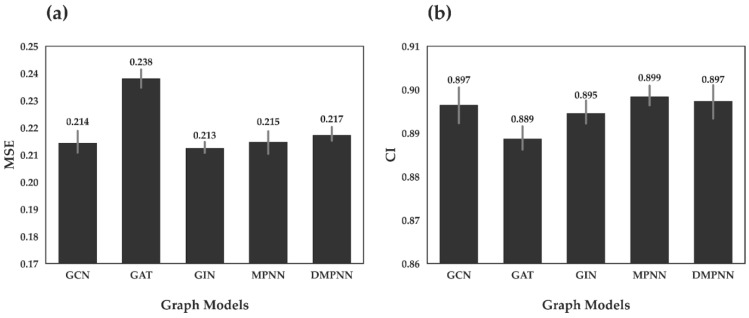
Binding affinity prediction results on the Davis test dataset. (**a**) Mean Squared Error (MSE); (**b**) Concordance Index (CI). We compared five GNN variants: the graph convolutional neural network (GCN), graph attention network (GAT), graph isomorphism network (GIN), message-passing neural network (MPNN), and direct message-passing neural network (DMPNN). The bars are the mean of ten models and the grey lines are error bars indicating the standard deviation.

**Figure 3 biomedicines-11-00067-f003:**
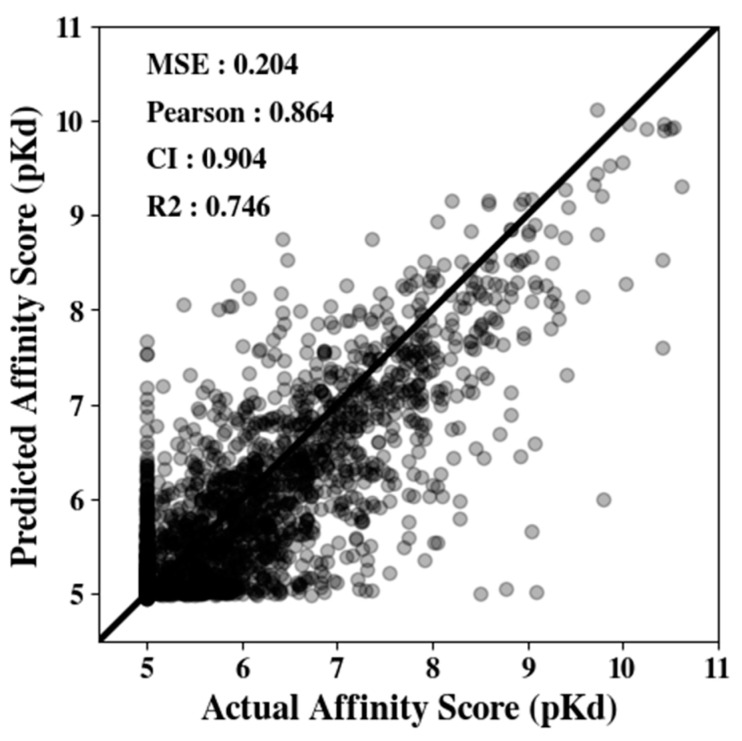
Prediction from the GraphATT-DTA model on the Davis testing data. The scatterplot shows the trend of the predicted affinity values vs. actual affinity values (pKd). The black line represents *y* = *x*. CI, concordance index; MSE, mean-squared error; Pearson, Pearson correlation; R2, r squared.

**Figure 4 biomedicines-11-00067-f004:**
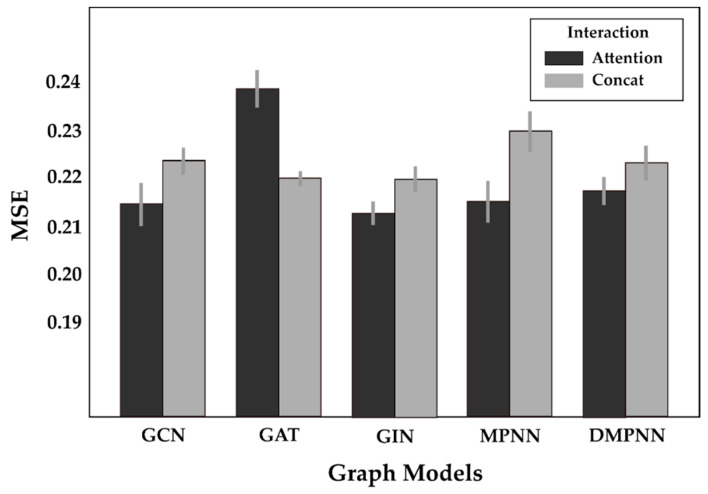
Ablation study on the Davis dataset.

**Figure 5 biomedicines-11-00067-f005:**
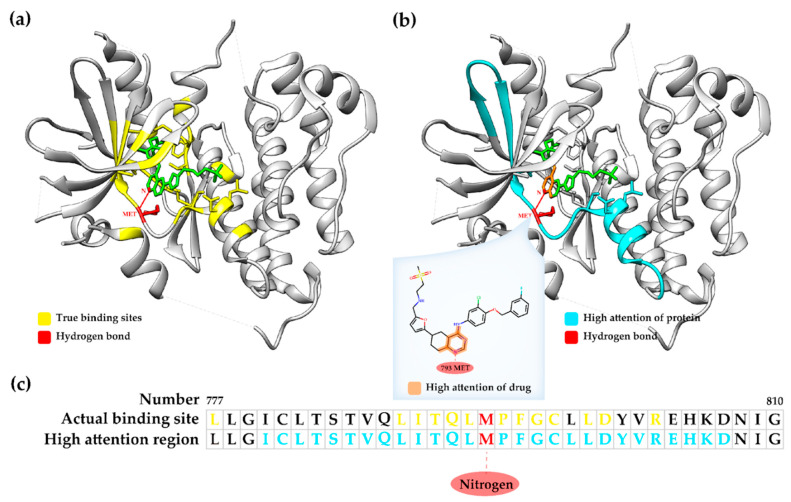
Visualization of drug-target binding affinities with significant regions: (**a**) complex compound Lapatinib and gene EGFR (PDB ID: 1XKK), where the ligand is colored green, true binding sites are yellow, the hydrogen bond is red; (**b**) complex 1XKK and visualization of substructure significance, where the high attention of drug is orange, high attention of protein is light blue; (**c**) visualization of sub-sequence significance.

**Table 1 biomedicines-11-00067-t001:** Datasets used for model training, validation, and testing.

Dataset	Davis	BindingDB
Proteins	442	509
Compounds	68	4,076
Interactions	30,056	14,505
Training	25,046	—
Testing	5010	14,505

**Table 2 biomedicines-11-00067-t002:** Input data representations: Compound atom features.

Feature	Dimension
One hot encoding of the atom element	44
One hot encoding of the degree of the atom in the molecule	11
One hot encoding of the total number of H bonds to the atom	11
One hot encoding of the number of implicit H bonds to the atom	11
Whether the atom is aromatic	1
All	78

**Table 3 biomedicines-11-00067-t003:** Input data representations: Compound bond features.

Feature	Dimension
One hot encoding of bond type	4
One hot encoding of bond conjugating	2
One hot encoding of whether the bond is part of a ring	2
One hot encoding of the stereo	6
All	14

**Table 4 biomedicines-11-00067-t004:** Graph neural network variants used for drug embedding matrix generation.

Model	Message Passing Function	Update Function
GCN	$m_{v}^{\left(t+1\right)}=\underset{w\in N\left(v\right)\cup^{\;}\left\{v\right\}}{\sum }\frac{1}{c_{\mathrm{wv}}}h_{w}^{\left(t\right)}$ $c_{\mathrm{wv}}=\frac{1}{\sqrt{\left|N\left(v\right)\right|\left|N\left(w\right)\right|}}$	$h_{v}^{\left(t+1\right)}=\sigma \left(m_{v}^{\left(t+1\right)}W^{t}\right)$
GAT	$m_{v}^{t+1}=\sigma \left(\underset{w\in N\left(v\right)\cup^{\;}\left\{v\right\}}{\sum }\alpha_{\mathrm{vw}}W^{\left(t\right)}h_{w}^{\left(t\right)}\right)$, $\alpha_{\mathrm{vw}}={\;\mathrm{softmax}}_{v}\left(e_{\mathrm{vw}}\right)$, $e_{\mathrm{vw}}=\mathrm{LeakyReLU}\left({\mathrm{Wh}}_{v},{\mathrm{Wh}}_{w}\right)$	$h_{v}^{\left(t+1\right)}={\|}_{k=1}^{K}m_{v}^{t+1}$, where ‖ is concatenation.
GIN	$m_{v}^{t+1}$ = $\underset{w\in N\left(v\right)}{\sum }\mathrm{MLP}\left(h_{w}^{\left(t\right)}\right)$	$h_{v}^{\left(t+1\right)}=\mathrm{MLP}\left(h_{v}^{\left(t\right)}+m_{v}^{t+1}\right)$
MPNN	$m_{v}^{t+1}$ = $\underset{w\in N\left(v\right)}{\sum }A\left(e_{\mathrm{vw}}\right){\;h}_{w}^{t}$	$h_{v}^{t+1}=\mathrm{GRU}\left(h_{v}^{t},m_{v}^{t+1}\right)$
DMPNN	$m_{vw}^{t+1}$ = $\underset{k\in \left\{N\left(v\right)\setminus w\right\}}{\sum }h_{\mathrm{kv}}^{t}$	$h_{vw}^{t+1}=\;\tau \left(h_{\mathrm{vw}}^{0}+W_{m}m_{\mathrm{vw}}^{t+1}\right)$

Notes: DMPNN, directed message-passing neural network; GAT, graph attention network; GCN, graph convolutional neural network; GIN, graph isomorphism network; MPNN, message-passing neural network.

**Table 5 biomedicines-11-00067-t005:** Hyperparameters for the GraphATT-DTA model.

Hyperparameter	Setting
Graph neural network layers	2 or 3 or 5
K size	8
Epoch	1000
Batch size	32
Learning rate	0.0005
Optimizer	Adam
Weight decay	0.00001
Embedding size of a feature	128
Fully connected layers	(1024, 512, 1)
Dropout	0.1 or 0.2
Early stopping	30

**Table 6 biomedicines-11-00067-t006:** GraphATT-DTA prediction performance on the Davis testing dataset vs. baseline models.

Models	Protein	Compound	Interaction	Davis MSE	Davis CI
DeepDTA	1D CNN	1D CNN	Concat	0.245	0.886
GraphDTA	1D CNN	GIN	Concat	0.229	0.890
DeepAffinity	RNN–CNN	RNN–CNN	Joint attention	0.302	0.870
ML-DTI	1D CNN	1D CNN	Mutual learning	0.222	0.891
HyperAttentionDTI	1D CNN	1D CNN	Hyperattention	0.233	0.876
FusionDTA	BI-LSTM	BI-LSTM	Fusion layer	0.203	0.911
GraphATT-DTA	1D CNN	MPNN	Interaction learning	0.204	0.904

Notes: BI-LSTM, bidirectional long short-term memory; CI, concordance index; CNN, convolutional neural network; GCN, graph convolutional neural network; GIN, graph isomorphism network; MSE, mean squared error; RNN, recurrent neural network.

**Table 7 biomedicines-11-00067-t007:** Prediction performance over the BindingDB external test set of GraphATT-DTA and baseline models.

Models	Interaction	Early Stopping	BindingDB MSE	BindingDB CI
DeepDTA	Concat	O	1.618	0.646
GraphDTA	Concat	X	2.13	0.62
DeepAffinity	Joint attention	X	2.188	0.574
ML-DTI	Mutual learning	O	1.580	0.704
HyperAttentionDTI	Hyperattention	O	1.514	0.656
FusionDTA	Fusion layer	X	1.970	0.567
GraphATT-DTA	Interaction learning	O	1.582	0.651

Notes: CI, concordance index; MSE, mean-squared error.

## Data Availability

Not applicable.

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
