# Peer review of "GraphATT-DTA: Attention-Based Novel Representation of Interaction to Predict Drug-Target Binding Affinity"

_biomedicines, 2022, doi:10.3390/biomedicines11010067_

Round 1

Reviewer 1 Report

The authors present an attention based method to predict the binding affinity of drugs and targets. The idea is interesting but there are several major problems that have to be addressed:

1. The authors do not clearly define what attention graphs are. Please provide a clear definition in the introduction or the methods.

2. The training and testing is only done on kinases. This in itself is not a major flaw but the authors have to emphasize it and, preferably, discuss whether it can be generalized to other enzymes or targets.

3. Some parts, in particular Section 3.5, require thorough revision for grammar and typos.

4. Figure 2, in the caption - what does it mean "Performance of various GNN methods with 10"?

5. Table 6 and 7 - the results are marginally better than most methods and in Table 6 slightly worse than Fusion DTA. Are there any other advantages such as run time?

Reviewer 2 Report

“Attention”, “Substructure of drugs” and “subsequence of proteins”: please provide additional explanations for the specific meanings in this paper.

How is the dynamics and deformability of the drug and protein handled?

I cannot comment on the details of the Methods section. This is not in this reviewer’s expertise.

The writing in the paper must be improved throughout. I suggest obtaining help from a native English speaker/writer with an interest in this field.

For the authors consideration:

Do different methods for determining DTA give better or worse performance for particular classes of drugs or classes of proteins? It seems that these data must be embedded in the analyses.

Is there a test dataset with several proteins with mutations that affect drug affinity? Such a test might provide additional information about computational DTA tools.

This reviewer imagines the following analysis. Do an in silico protein mutagenic and drug modification study to better understand how drugs interact with proteins. If possible, include analyses of protein and drug dynamics and solvent interactions.  

Minor points:

Line 27: suggest using “because” in place of “since”. Suggest: replace all “since” in manuscript with “because”.

Authors should learn the formal use of “which” and “that”: mostly, in scientific writing, “that” is most appropriate. Suggest: replace all “which” in manuscript with “that”.

Lines 37, 38: wording is awkward

Line 53: “informative information” is not good English

Throughout, the manuscript is poorly written. The manuscript will have a greater impact if it is easier to read.

Round 2

Reviewer 1 Report

The authors addressed most of the concerns raised by the two reviewers but there are still several issues that have to be addressed:

1. There are numerous formatting issues (maybe it is just on my computer). For example: Fig. 4 caption is missing, some section titles - e.g. 3.4, 3.5 etc. repeat twice, and many lines - e.g. 272-273 among many others, are garbled in a way that makes it impossible to read.

2. I am not sure I understand the reply to point 5 re. the performance of the method with respect to FusionDTA. I quote:

We interpret it as the result of FusionDTA's overfitted learning for the
Davis datasets. In machine learning studies, the performance reported using the external dataset is regarded as a more generalized performance with higher confidence. Thus, we can conclude that the proposed model incorporating the attention mechanism between a subsequence of the target protein and a substructure of the compound demonstrated superior performance in terms of prediction compared to FusionDTA.

--- (end quote).

Is it a statement that the authors can make with confidence? What data do you have to support this claim?
